# Can Active Aerobic Exercise Reduce the Risk of Cardiovascular Disease in Prehypertensive Elderly Women by Improving HDL Cholesterol and Inflammatory Markers?

**DOI:** 10.3390/ijerph17165910

**Published:** 2020-08-14

**Authors:** Nayoung Ahn, Kijin Kim

**Affiliations:** Department of Physical Education, Keimyung University, Daegu 42601, Korea; nyahn13@kmu.ac.kr

**Keywords:** hypertension, exercise, inflammation, cardiovascular disease, physical fitness

## Abstract

This study aims to verify the efficacy of exercise programs designed to prevent and treat hypertension-induced cardiovascular disease (CVD) by analyzing the effects of a 6-month active aerobic exercise program, administered to prehypertensive elderly women, on reducing the risk of developing CVD by enhancing their physical fitness level and improving the detailed markers of high-density lipoprotein cholesterol (HDL-C) and inflammatory markers. We assigned the elderly women (≥65 years) recruited into normal blood pressure (120–129/80–84; NBP, *n* = 18) and high-normal blood pressure (130–139/85–89; HNBP, *n* = 12) groups according to the European guidelines for the management of arterial hypertension. The exercise program was made up of combined workouts of elastic band resistance exercise and aerobics with dance music. The program took place three times a week for six months, with each session lasting 60 min. We measured pre- and post-intervention body composition, blood pressure, physical fitness level, blood lipids profile, HDL-C, SAA, TNF-α, IL-6, IL-4, IL-15, CRP, and HSP70 and calculated the Framingham risk scores for comparison. A significant post-intervention reduction in the mean systolic blood pressure (SBP) was observed in the HNBP group (*p* < 0.001), with significant increase in HDL-C (*p* < 0.01) and significant decrease in serum amyloid A (SAA) concentration (*p* < 0.01). A significant improvement in physical fitness factors such as physical efficiency index (PEI) was also observed in the HNBP group (*p* < 0.05). The post-intervention TNF-α, IL-6, and SAA concentrations were more significantly lower in the HNBP than in the NBP group (*p* < 0.05). Compared to the baseline values, a significant decrease in SAA concentration (*p* < 0.01) and significant increase in HSP70 concentration (*p* < 0.001) were observed in the HNBP group. The HNBP group’s 10-year CVD risk was also significantly reduced (*p* < 0.05). The pre–post differences in SBP and DBP were significantly correlated with those in the anti-inflammatory markers IL-4 and IL-15 (*p* < 0.01). In conclusion, the 6-month active aerobic exercise program of moderate intensity administered to prehypertensive elderly women (≥65 years) had the effect of reducing the 10-year CVD risk through a substantial reduction in SBP, overall physical fitness improvement, increase in HDL-C, decrease in SAA concentration, and substantial decrease in inflammatory biomarkers. It was also confirmed that an increase in anti-inflammatory markers, which showed a small range of increase with respect to the decrease in blood pressure, may have a major effect.

## 1. Introduction

Responsible for over 50% of all deaths caused by cardiovascular disease (CVD), hypertension is a major global public health risk factor [1]. The Seventh Report of the Joint National Committee on Prevention, Detection, and Treatment of High Blood Pressure (JNC7) defines prehypertension as systolic blood pressure (SBP) readings of 120–139 mmHg or diastolic blood pressure (DBP) of 80–89 mmHg [2]. Prevalence of prehypertension is high among adults, with the risk of going on to develop chronic hypertension increasing with age [3,4]. In particular, hypertensive CVD causes atherosclerotic vascular lesions and is a leading cause of death worldwide [5,6]. The European guidelines for the management of arterial hypertension [7] subdivided the range of prehypertension into normal blood pressure (120–129/80–84; NBP) and high-normal blood pressure (130–139/85–89; HNBP). HNBP carries CVD risk factors and is associated with a very high prevalence of CVD and coronary artery disease (CAD) [8].

In the prehypertensive stage, pathological conditions including CVD cause high-density lipoprotein cholesterol (HDL-C) to undergo structural changes, thereby impairing its atheroprotective and anti-inflammatory functions, which in turn can impair its protective function by increasing the level of dysfunctional HDL-C [9]. Increased serum amyloid A (SAA), a major acute-phase protein acting as a biomarker of dysfunctional HDL-C, makes HDL-C particles dysfunctional by binding and circulating with them, impairing HDL’s antioxidant activity [10]. SAA enrichment in HDL-C molecules acts as a systemic inflammatory marker indicative of an ongoing inflammatory process, during which SAA increases rapidly. High levels of SAA are shown in the CVD high risk group [9], and high enrichment of HDL-C with SAA is closely associated with CVD prevalence and mortality [11]. Furthermore, CVD risk in hypertension patients is more closely associated with the levels of apolipoprotein, which plays a crucial role in the protective functions of HDL, than with the quantitative levels of HDL-C [12], whereby CVD patients have low concentrations of serum apolipoprotein A1 (ApoA1) [13]. Hypertensive CVD patients also show elevated plasma concentrations of tumor necrosis factor-α (TNF-α), one of the typical inflammatory mediators, and elevated levels of C-reactive protein (CRP) with monocyte and macrophage infiltration and accumulation in artery walls [14]. As examined above, progression from prehypertension to hypertension increases not only biomarkers of metabolic disorders, but also cellular inflammatory markers [15,16,17]. Moreover, progression to chronic inflammation and CVD, including atherosclerosis, triggered by endothelial cell activation [18], is characterized by accompanying complications [19,20].

The most effective intervention for prevention of hypertension in the prehypertension, in particular HNBP stage is administering an intensive lifestyle change program based on a low-salt diet and regular exercise aimed at achieving and maintaining blood pressure within an optimal range [1]. Studies on the prevention of hypertension and related complications [21,22,23,24,25,26] attach great importance to exercise programs. It has been reported that the level of fitness enhanced through exercise program has an important effect on blood pressure control [21,27] and greatly contributes to reducing CVD mortality [22,23]. Epidemiological investigation verified the effects of regular aerobic exercise in preventing and treating hypertension and reducing CV risk and mortality. A meta-analysis of randomized controlled trials (RCTs) confirmed that aerobic endurance training, dynamic resistance training, and isometric training had the effects of reducing resting SBP and DBP by 3.5/2.5, 1.8/3.2, and 10.9/6.2 mmHg, respectively [28]. However, these findings are not consistent with the findings of other studies that suggest exercise capacity is not related to blood pressure [29] and that physical fitness level is associated with obesity, but not with blood pressure [30].

Studies investigating the association between exercise and CVD risk factors reported that exercise training has a direct effect on vascular function and structure [30], but changes in traditional cardiovascular risk factors are not correlated with exercise in terms of arterial function and health [31]. However, in a study in which patients with chronic obstructive pulmonary disease (COPD) were treated with aerobic exercise, increase in ApoA1 concentration was observed [13]. In a study analyzing the effects of long-term physical activity, HDL-C and ApoA1 levels were increased by 10 and 11%, respectively, but no change was observed in the level of SAA component of HDL-C [32]. Regular exercise enhances cell recovery and antioxidant effects. In particular, exercise-induced overexpression of heat shock protein 70 (HSP70) inhibits tissue damage by inhibiting mitochondrial damage [33,34,35], and also inhibits apoptosis of cardiac muscle cells and protects cells [36]. Furthermore, the expression level of superoxide dismutase (SOD), which acts as regulator of oxidative stress suppression in the cardiovascular system, is increased through regular aerobic exercise [37]. A study on exercise treatment of aging rats found that exercise treatment significantly increased the activities of total SOD, mitochondrial SOD, cytoplasmic SOD, and extracellular SOD [38].

However, in the process of analyzing the exercise training effects on free radicals and inflammatory markers in the prehypertension stage, detailed mechanisms related to the effects of treating CVD-related risk factors including functional markers of HDL-C and inflammatory biomarkers are still unclear. Therefore, it is necessary to analyze exercise programs with regard to the symptoms of prehypertension, physical fitness level, inflammatory markers, and heart disease predictors.

Considering the foregoing, this study aims to verify the efficacy of exercise programs designed to prevent and treat hypertension-induced CVD by analyzing the effects of a 6-month active aerobic exercise program, administered to prehypertensive elderly women, on reducing the CVD risk by enhancing their fitness level and improving the detailed markers of HDL-C and inflammatory markers.

## 2. Methods

### 2.1. Subjects

Elderly women (≥65 years) were randomly selected for blood pressure (BP) measurement and were assigned to either the normal blood pressure group (120–129/80–84; NBP, *n* = 18) or the high-normal blood pressure group (130–139/85–89; HNBP, *n* = 12) according to the European guidelines for the management of arterial hypertension [7] in Table 1. The study subjects were selected from among the members of the Elderly Health Promotion Center (out of 1000 members) in Daegu, South Korea, who viewed the notice and voluntarily agreed to participate. However, it was composed of subjects confirmed by the doctor’s diagnosis as having no special diseases except blood pressure. Therefore, the limitation of this study is that the random selection method cannot be considered to be applied in the process of selecting a perfect subject. A total of 40 participants were enrolled (20 in each group), but 30 participants completed the program, with 10 dropouts during the 6-month experimental period (NBP group: *n* = 2, HNBP group: *n* = 8). In the process of selecting candidates among subjects who were allowed to participate through the doctor’s diagnosis and randomly classifying them into two groups, the limitation of the sample size was considered as the limitation of this study. We obtained approval for the recruitment and exercise treatment experiment from the Institutional Review Board (IRB) of Keimyung University (40525-201706-BR-27-02).

### 2.2. Active Aerobic Exercise Program

The exercise program was made up of combined workouts of elastic band resistance exercise and aerobic dance with music. The program took place three times a week for six months, with each session lasting 60 min. Exercise intensity was set in compliance with the recommendations for seniors on the Borg’s rating of perceived exertion (RPE) scale proposed by the American College of Sports Medicine (ACSM) [39]. Prior to beginning the exercise program, the participants were given a detailed explanation of RPE, and the exercise intensity was monitored during the session through continuous communication. The intensity of effort was set at 11–13 (light to somewhat hard) from week 1 to week 12, and 13–15 (somewhat hard to hard) from week 13 to week 24. The elastic band color was changed every week to build resistance in tune with individual exercise capacity and exercise intensity was progressed by increasing the number of repetitions. The yellow resistance band initially used was replaced by the red band after four weeks, subject to individual exercise capacity. Resistance band exercise for the upper and lower extremities consisted of 10 repetitions of three sets of seven movements (seated rows, elbow flexion, archery pull for posterior shoulder, hip flexion, hip extension, long-sitting ankle plantar flexion), which last 20 min in total with 1-min break between sets.

An aerobic dance workout to music was sustained for 20 min, with care taken to keep the heart rate (HR) within the range of individual target HR ± 5% by setting the target HR at 65–75% HRmax (moderate intensity) using the Karvonen formula [(HR max−HR rest) × exercise intensity (65–75%) + HR rest]. The post-exercise heart rate was monitored by measuring the heart rate of the radial and carotid arteries (HR/min = HR/15 s × 4). The 10-min warm-up and cool-down exercises consisted mainly of stretching. Adjustment of exercise intensity was applied in the range of 65–75% HRmax as a whole, and was maintained in the range of 65–70% HRmax in the first half and 70–75% HRmax in the second half. Control of this intensity was monitored using heart rate and RPE as described above.

### 2.3. Body Composition and Function Measurements

Body height (H) and weight (W) were measured using a stadiometer and InBody 3.0 (Biospace, Korea), respectively. The body mass index (BMI) was calculated using the standard formula: BMI = W/H^2^ (kg/m^2^). The senior fitness test of Rikli and Jones was used to measure the physical function and life fitness of the elderly [40]. The Rikli and Jones Senior Fitness Test (SFT) was administered to assess the participants’ physical and functional fitness [40]. The SFT assessment items were 30-s chair stand (number of sit-to-stand during 30 s), 30-s arm curl (number of arm curl repetitions performed with a 2-kg dumbbell during 30 s), chair sit and reach (distance between the fingertips and the toes while reaching forward towards toes without bending the knees, and back scratch (distance between the middle fingers of both hands with the arms placed around the back). In addition, the 10-feet (~3 m) walk test used in Gill’s study was administered to measure the fast walking speed, with the participant rising from a chair, walking 3 m, and returning to the chair. Physical efficiency index (PEI) was calculated using the formula [PEI = (100 × test duration in seconds)/(2 × HR Sum in the recovery periods)]. As body composition variables, body fat percentage and waist-to-hip ratio (WHR) were analyzed as well.

### 2.4. Blood Collection and Biochemical Analysis

A fasting (12 h) blood sample (10 mL) was collected from each participant through the brachial vein. The collected blood was heparin-treated and centrifuged at 3000 rpm for 10 min, and TC, TG, and LDL-C were analyzed with an enzymatic method using an automatic analyzer (Auto-analyzer Hitachi 7150, Hitachi Ltd., Tokyo, Japan). HDL-C was measured by an enzymatic method using the supernatant after precipitation HDL-C particles using a precipitating agent. Blood glucose was measured using an automatic blood glucose analyzer (YSI 2300, Marshall Scientific, Hampton, NH, USA), and insulin was measured using an insulin ELISA kit (Mercodia, Uppsala, Sweden). Homeostatic model assessment for insulin resistance (HOMA IR), an index of insulin resistance, was calculated with the formula described by Matthews et al. [41]: fasting serum insulin (μL/mL) × fasting plasma glucose (mmol/L)/22.5.

### 2.5. Analysis of Pro-Inflammatory and Anti-Inflammatory Markers

Tumor necrosis factor-α (TNF-α) and IL-6 were analyzed using Luminex technology (PPX-056_MXCE4A4, 191482000). Enzyme-linked immunosorbent assay (ELISA) was used to analyze CRP (R&D DCRP00, P183393 kit), SAA (Elabscience ab100635, KA0528 kit), ApoA-1(Abcam ab108804, GR3188830-3 kit), SOD2(Abnova KA0528 kit), and HSP70 (Enzo ADI-EKS-715, 02201818A kit).

### 2.6. Calculation of the 10-Year Risk of Heart Disease

Using the measured values of the corresponding test variables such as gender, age, TC, HDL-C, LDL-Cl, SBP, DBP, diabetes, and smoking status, we calculated a 10-year risk of heart disease with the Framingham risk score [42]. First, after dividing the age of women into nine groups and LDL-C, TC, HDL-C, SBP, and DBP into five groups each, we assigned a risk score to each group. Diabetes, hypertension, and smoking status were treated as binary (yes/no) variables and scores were given accordingly. The scores assigned to the test variables were taken into account in six stages: age in stage 1, either LDL-C or TC in stage 2, HDL-C in stage 3, SBP and DBP in stage 4, diabetics in stage 5, and smoking status in stage 6. Different scores reflecting LDL-C or TC were assigned across the stages in all variables. The 10-year risk of heart disease was calculated based on the sums of the scores reflecting LDL-C or TC.

### 2.7. Statistical Analysis

Statistical analysis was performed using SPSS version 24.0 (IBM SPSS Statistics, Chicago, IL, USA). The independent *t*-test was used to evaluate intergroup differences in general physical characteristics. Two-way repeated ANOVA was performed to evaluate the significant differences between the groups and the periods, followed by a post-hoc test; when significant differences were found: Mann–Whitney U test between groups by period and the Wilcoxon test between periods by group. Pearson’s correlation was performed to analyze the correlations between the changed post-intervention values of metabolic syndrome predictors and major items. All data were described as mean ± standard deviation, and the statistical significance level was set at *p* < 0.05.

## 3. Results

### 3.1. Body Composition

No significant intergroup and intragroup (pre–post) differences were observed in body weight in Table 2. The mean BMI fell under the obesity class by Korean criteria, with the post-intervention value of body fat percentage (%fat) significantly higher (*p* < 0.05) in the HNBP group than in the NBP group as well as pre-intervention value of %fat within the HNBP group.

### 3.2. CVD Risk Factors

SBP decreased significantly (*p* < 0.001) by 2.1 mmHg in the HNBP group after the exercise program, but DBP remained unchanged. HDL-C increased significantly (*p* < 0.01) and TG concentration decreased significantly (*p* < 0.01) in the HNBP group after the exercise program in Table 2. In particular, the pre-intervention values were significantly higher (*p* < 0.001) in the HNBP group compared with the NBP group. Significantly higher post-intervention values were observed in the levels of blood insulin (*p* < 0.05) and glucose (*p* < 0.01) in the HNBP group, but not in HOMA IR. Compared with the NBP group, however, the HNBP group showed significantly higher post-intervention levels of glucose (*p* < 0.01) and HOMA IR (*p* < 0.05).

### 3.3. Physical Fitness

No intergroup differences were observed in physical fitness factors except for arm curl. In the intragroup pre–post comparison, the HNBP group showed significant improvements in the chair sit and reach (*p* < 0.05), 30-s chair stand (*p* < 0.001), arm curl (*p* < 0.01), and PEI (*p* < 0.05), as did the NBP group in back scratch (*p* < 0.01), 30-s chair stand (*p* < 0.001), and arm curl (*p* < 0.01) in Table 3.

### 3.4. Pro-Inflammatory and Anti-Inflammatory Markers

Analysis of pro-inflammatory markers revealed significantly greater decreases in the TNF-α, IL-6, and SAA levels in the HNBP compare with the NBP group (*p* < 0.05) after the 6-month exercise program in Table 4. A particularly significant (*p* < 0.01) post-intervention decrease in SAA was observed in the HNBP. In contrast, the NBP did not show any positive effects of exercise. Analysis of anti-inflammation markers revealed a significant increase in the HSP70 level (*p* < 0.001) in the HNBP after the 6-month exercise program. In the NBP group as well, significant increases were observed in ApoA1 (*p* < 0.05), SOD2 (*p* < 0.01), and HSP70 (*p* < 0.05).

### 3.5. Estimation of 10-Year Risk of Coronary Heart Disease

In both pre- and post-intervention estimations, the HNBP group showed a significantly (*p* < 0.001) higher 10-year risk of coronary heart disease than the NBP group in Table 5. However, the intragroup pre–post comparison revealed a significant decrease (*p* < 0.05) after the exercise program. In cardiovascular age as well, the HNBP group was significantly (*p* < 0.001) higher than the NBP group.

### 3.6. Correlations between Blood Pressure and Anti-Inflammation Markers

Analysis of major factors that showed significant pre–post differences in SBP and DBP identified the anti-inflammatory factors, IL-4 and IL-15, as factors with the highest correlation coefficients in Figure 1. In particular, the changed amount of IL-15 was significantly associated with those of SBP and DBP, yielding correlation coefficients of −0.734 and −0.628, respectively.

## 4. Discussion

In this study, SBP decreased by 2.1 mmHg in the HNBP group after the 6-month exercise program, and significant improvements were observed across the physical fitness factors after the 6-month intervention program, verifying the positive effects of aerobic exercise programs on reducing blood pressure [43,44]. As the most basic step in the factor analysis process related to the post-intervention reduction in post-intervention SBP, which is the typical phenotypic marker or vascular function, changes in the levels of HDL-C and SAA were monitored based on blood lipid variables. Circulating SAA and HDL-C levels are inversely correlated in patients with hypertension and metabolic syndrome, and elevated SAA levels actively inhibit HDL-C’s antioxidant activity [45]. A study analyzing the association between the CVD mortality rate and SAA level in CVD patients reported that all CVD risk factors were closely associated with elevated SAA levels [46]. In this study, the post-intervention SAA level in the HNBP group was significantly reduced compared with the pre-intervention value. In particular, the HDL-C level also increased significantly after exercise, which allows the assumption that the active aerobic exercise program for the prehypertensive elderly had a positive effect on HDL-C’s vaso-protective function. Moreover, in the efficacy analysis process for the prevention and treatment of hypertension and CVD, it was confirmed that SAA plays a more important role, compared with the functional role played by HDL-C, through the changes in the ratio of SAA concentration relative to HDL-C level instead of the absolute levels of HDL-C and SAA. Many studies have advanced the view that exercise improves metabolic and immune functions by improving blood lipids and inflammatory markers. However, the mechanisms mediating the effects of vascular endothelial function and tissue cells are still unclear, and research analyzing the reciprocal effects and close association between the structural change in dysfunctional HDL-C and cardiovascular tissues is still in its infancy. Nevertheless, the positive effect of exercise training will contribute to preventing CVD and metabolic syndrome and reducing the accompanying impairment of physical functions secondary to CVD by reducing the SAA level corresponding to dysfunctional HDL-C, as verified in this study.

The antioxidant function of HDL-C comes into play in close interaction of enzymes, such as Lp-PLA2 (lipoprotein-associated phospholipase A2), PON1 (paraoxonase1), LCAT (lecithin cholesterol-acyltransferase) and GSPx (glutathione selenoperoxidase), and ApoA1 [47,48,49]. ApoA1 has the strongest antioxidant effect among all Apos in the prevention of LDL oxidation [48], removing oxidized phospholipids from LDL and arterial wall cells [49]. This allows the estimation of CAD according to HDL’s pro-inflammatory and anti-inflammatory properties [50], and exercise was expected to reduce the expression of inflammation-related genes by improving modified ApoA1, thus contributing to a chronic inflammatory state transformation CAD reduction [51]. Hypertensive patients have high levels of cellular inflammatory markers as well as cell adhesion molecules [15,16,17]. However, exercise reduces the levels of circulating IL-6, CRP, and TNF-α [52], and it has also been reported that physical activity in normotensive people is negatively correlated with CRP, fibrinogen, and white blood cell count [53] as well as inflammatory markers including TNF-α and IL-6. In this study, the HNBP group showed a significant decrease in the post-intervention levels of TNF-α, Il-6, and SAA, but these inflammatory markers (CRP, TNF-α, Il-6, and SAA) showed no significant change in the NBP group, with only the level of ApoA1 significantly increased. It can be said that these differences between groups are difficult to accept as clear results. Therefore, it is thought that continuous research is required to further verification this part. From the finding that exercise training has a direct effect on vascular function and structure in a study analyzing the association between exercise and CVD risk factors [54] and that changes in the traditional cardiovascular risk factors are independent of exercise-induced adaptation with regard to arterial function and health [31], it could be inferred that there is no clear evidence of the correlation between the expression of inflammatory markers and the metabolic function in the course of analyzing the mechanisms of exercise-induced treatment of hypertension and CVD-related risk factors caused by exercise.

Exercise-induced high serum HSP70 level protects the production of atherosclerosis [55], and long-term exercise adaptation maintains a high level of HSP70 protein expression in aorta and myocardium as well as skeletal muscle, leading to higher resistance to myocardial cell stress [56]. According to a study of antioxidant effects of exercise in elderly men (mean age: 62; *n* = 18), a 3-month swimming and running exercise program resulted in an increase in plasma SOD2 level [57], and a regular aerobic exercise group showed higher SOD2 level than the sedentary control group [58]. Moreover, HSP70 protein increased in leukocytes in male marathon runners after a half marathon [59], 120-min of rapid cycling in a 22-year-old man led to increase in plasma HSP72 level [60] and increase in serum HSP72 level after a 60-min treadmill exercise [61]. Aerobic and resistance exercise training in the elderly lowers the levels of inflammatory markers such as TNF-α, IL-6, and CRP [62,63,64,65] and increases myokine such as Il-15, thus playing an anti-inflammatory role [66]. In the current study, SOD2 increased significantly after exercise only in the NBP group, but HSP70 increased significantly in both groups, which verified that an exercise program for elderly women can have an antioxidant effect. However, although exercise had the effect of triggering the activity of HSP70 and SOD2 protein in this study, it could not demonstrate the direct effect on cell damage including apoptosis. Therefore, to verify more clearly the effect of preventing and treating hypertension, further research is necessary to analyze the effects of exercise programs reflecting age-dependent appropriate exercise intensity and duration in terms of apoptosis-related factors and cell recovery mechanisms. Furthermore, in view of the lack of change in SOD2 concentration in the HNBP group after exercise, a study will have to analyze the effect of detailed exercise programs considering the participants’ general characteristics.

TNF-α, a typical inflammatory cytokine, was found to be lower in physically active people than in sedentary people, from which it can be inferred that exercise is associated with the control of inflammatory cytokines, and aerobic and muscle strengthening activities were found to decrease the TNF-α level in older adults [62]. However, a high-intensity aerobic exercise and moderate-intensity muscle strength training intervention in patients with acute heart failure and CAD had no effect on the plasma concentration of TNF-α [67]. Likewise, a study with older adults [68] reported that an aerobic and resistance exercise program had no effect on the TNF-α and IL-6 levels. Taking these results together, there is no clear evidence that inflammatory cytokines vary depending on exercise type, intensity, and duration.

In the current study, the 10-year CVD risk and cardiovascular age were estimated to be significantly higher (*p* < 0.001) in the HNBP group compared with the NBP group, but significantly lower (*p* < 0.05) relative to the baseline value. This leads to the assumption that high blood pressure is closely associated with the prevalence of CVD and hypertension is a risk factor for increasing cardiovascular age, demonstrating that exercise can reduce the risk of CVD prevalence even in hypertensive people. The Framingham Risk Score (FRS) is the sum of the scores calculated from demographic and health-related variables such as gender, age, LDL cholesterol or total cholesterol, HDL-C, SBP, DBP, diabetes, and smoking status as described in the Framingham heart study. It is most widely known as an assessment tool that can predict CVD risk accurately and efficiently by complexly evaluating multiple factors [69]. Although several studies have verified the validity of the FRS [70], its over- or underestimating tendency for specific diseases is still controversial [71]. Moreover, only a limited number of studies has investigated the pre–post differences in the FRS and effects of exercise training treatment. It has also been reported that exercise activity decreases SBP and DBP and improves blood lipid variables [72], and clear improvements in blood pressure and blood lipid variables act as important factors for reducing the FRS [73]. However, as shown in the results of the current study, the exercise program had an effect of improving FRS, but only SBP was significantly reduced among the related detailed factors, which highlights the necessity for continuous study to investigate the overall effects of exercise programs.

In the study of exercise treatment in the elderly, one of the main reasons for the differences in the results regarding the changes in physical fitness, metabolic changes, and inflammatory and anti-inflammatory marker levels is setting the exercise intensity and duration. While most studies are conducted with moderate intensity exercise, no changes in metabolites and inflammatory markers are observed in some studies. According to research results, intense exercise reduces CVD risk factors [74], with high-intensity interval training (HIIT) proving more effective than moderate-intensity exercise in strengthening cardiovascular functions and physical fitness [75], and high- and moderate-intensity exercise in overweight and obesity groups being associated with improvement in cardiometabolic risk factors such as body fat percentage and maximal oxygen intake, which is a blood lipid variable [76]. This study analyzed major variables that have significant effects on the post-intervention changes in SBP and DBP, which are key phenotypic markers of vascular functions, and identified the anti-inflammatory factors, IL-4 and IL-15, as the variables with the highest correlation coefficients. In particular, the amount of change in IL-15 showed significant correlations with SBP and DBP (correlation coefficients: −0.734 and −0.628, respectively). This leads to the assumption that a small range of post-intervention reduction in anti-inflammatory markers has a positive effect on blood pressure reduction. However, no other variables were found to correlate with changes in IL-4 and IL-15 and pathways to better blood pressure. Therefore, in dealing with prehypertensive patients, in-depth studies need be conducted to support the importance of improving blood pressure by activating anti-inflammatory factors. However, it is considered that this study is limited to the elderly in a limited area and the application of the results is very limited considering the very small sample size, and further studies including more subjects are required.

## 5. Conclusions

As a result of a 6-month moderate-intensity active aerobic exercise program administered to prehypertensive elderly women (≥65 years), systolic blood pressure was significantly reduced and improvements in vascular function markers could be verified through the improvement of overall physical fitness, including cardiopulmonary endurance, increase in HDL-C, decrease in serum SAA concentration, and substantial decrease in inflammatory biomarkers. It was also verified that exercise training can greatly contribute to reducing 10-year cardiovascular disease risk and that an increase in anti-inflammatory markers, even a small-range increase, can have a great impact on reducing blood pressure.

## Figures and Tables

**Figure 1 ijerph-17-05910-f001:**
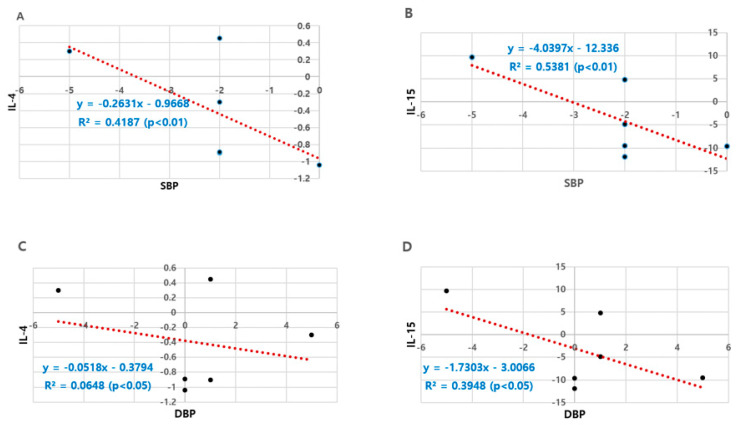
Correlation between blood pressure and anti-inflammation cytokines. (**A**) SBP vs. IL-4; (**B**) SBP vs. IL-15; (**C**) DBP vs. IL-4; (**D**) DBP vs. IL-15; R2: Pearson’s correlation coefficient.

**Table 1 ijerph-17-05910-t001:** Characteristic of subjects.

Variable	NBP (*n* = 18)	HNBP (*n* = 12)
Age (yr)	69.22	72.00
	4.14	3.81
Height (cm)	154.11	152.17
	4.35	5.31
Body weight (kg)	61.20	58.43
	9.36	7.30
BMI (kg/m^2^)	25.23	25.27
	3.87	2.53
%fat	34.06	37.08
	6.03	4.09
WHR	0.90	0.90
	0.07	0.09

Values are mean and standard deviation; NBP: normal blood pressure; HNBP: high-normal blood pressure; BMI: body mass index; WHR: waist hip ratio.

**Table 2 ijerph-17-05910-t002:** Change of body composition and cardiovascular disease risk factors after exercise.

Variable	NBP (*n* = 18)		HNBP (*n* = 12)		Source	*F*-Value	*p*-Value
	Pre	Post	Pre	Post			
***Body composition***							
Body weight (kg)	61.20	61.37	58.43	58.82	Time	0.015	0.902
	9.36	9.36	7.30	6.49	Groups	1.391	0.001
					T × G	0.002	0.961
BMI (kg/m^2^)	25.23	25.27	25.27	25.37	Time	0.005	0.946
	3.87	3.81	2.53	2.16	Groups	0.007	0.932
					T × G	0.001	0.973
%fat	34.06	33.56	37.08	38.97 *^,#^	Time	0.272	0.606
	6.03	6.06	4.09	3.83	Groups	8.160	0.008
					T × G	0.807	0.377
WHR	0.90	0.87 ^#^	0.90	0.87 ^#^	Time	13.440	0.001
	0.07	0.05	0.09	0.05	Groups	0.001	1.000
					T × G	0.001	1.000
***Cardiovascular disease risk factors***							
SBP (mmHg)	118.44	118.89	135.67 ***	133.50 ***^,###^	Time	2.033	0.165
	10.65	7.19	3.94	4.66	Groups	33.285	0.001
					T × G	4.673	0.039
DBP (mmHg)	77.44	78.89 ^#^	83.00 *	83.33	Time	2.621	0.117
	7.97	7.19	4.22	3.89	Groups	4.583	0.041
					T × G	1.024	0.320
TC (mg/dL)	178.56	193.11	182.00	189.33	Time	2.052	0.163
	48.80	40.36	47.17	48.69	Groups	0.001	0.991
					T × G	0.223	0.640
HDL-C (mg/dL)	54.67	55.44	51.17	57.67 ^##^	Time	7.681	0.010
	6.77	9.14	7.32	7.25	Groups	0.062	0.806
					T × G	4.749	0.038
LDL-C (mg/dL)	111.44	126.11	105.00	116.83	Time	4.880	0.036
	41.40	36.49	34.60	41.47	Groups	0.359	0.554
					T × G	0.056	0.815
TG (mg/dL)	97.67	104.78	157.33 ***	124.67 ^##^	Time	2.224	0.147
	37.59	43.48	39.04	37.04	Groups	10.922	0.003
					T × G	5.389	0.028
Insulin	7.56	8.14	6.30	9.00 ^#^	Time	5.692	0.024
	3.79	3.21	2.45	2.29	Groups	0.046	0.832
					T × G	2.345	0.137
Glucose	91.78	95.89	110.00 **	123.67 **^,##^	Time	25.080	0.001
	10.97	18.45	23.53	31.83	Groups	8.963	0.006
					T × G	7.246	0.012
HOMA IR	1.68	1.95	1.64	2.77 *	Time	9.791	0.004
	0.79	0.91	0.54	1.26	Groups	2.403	0.132
					T × G	3.714	0.064

Values are mean and standard deviation; NBP: normal blood pressure; HNBP: high-normal blood pressure; BMI: body mass index; WHR: waist hip ratio; SBP: systolic blood pressure; DBP: diastolic blood pressure; TC: total cholesterol; HDL-C: high density lipoprotein cholesterol; LDL-C: low density lipoprotein cholesterol; TG: triglyceride; HOMA IR: homeostatic model assessment insulin resistance; T: time; G: groups; * *p* < 0.05, ** *p* < 0.01, *** *p* < 0.001, vs. NBP group: Mann–Whitney test between groups; # *p* < 0.05, ## *p* < 0.01, ### *p* < 0.001 vs. pre: Wilcoxon test between time.

**Table 3 ijerph-17-05910-t003:** Change of physical fitness after exercise.

Variable	NBP (*n* = 18)		HNBP (*n* = 12)		Source	*F*-Value	*p*-Value
	Pre	Post	Pre	Post			
Chair sit and reach (cm)	21.88	21.26	17.45	22.50 ^#^	Time	5.531	0.027
	3.11	6.80	3.76	6.59	Groups	1.603	0.217
					T × G	4.190	0.051
Back scratch (Right, cm)	−7.02	−4.00 ^##^	−9.88	−7.25	Time	2.548	0.125
	6.63	3.53	7.86	4.63	Groups	3.000	0.097
					T × G	0.288	0.567
30 s chair stand (time)	16.56	25.33 ^###^	14.17	25.83 ^###^	Time	107.785	0.001
	2.75	5.86	1.53	4.57	Groups	1171.705	0.001
					T × G	2.152	0.154
Arm curl (Right, time)	25.38	31.00 ^##^	22.33 *	29.50 ^##^	Time	36.774	0.001
	3.72	4.50	3.85	2.61	Groups	3.653	0.067
					T × G	1.166	0.290
8 foot up-and-go (sec)	14.67	14.90	15.83	15.98 **	Time	0.190	0.666
	4.32	1.79	1.82	1.91	Groups	3.675	0.065
					T × G	0.008	0.929
PEI (Harvard step test)	99.09	114.57	94.24	103.76 ^#^	Time	3.813	0.065
	15.13	14.15	7.09	8.75	Groups	10.090	0.005
					T × G	0.130	0.722

Values are mean and standard deviation; NBP: normal blood pressure; HNBP: high-normal blood pressure; PEI: physical efficiency index; T: time; G: groups; * *p* < 0.05, ** *p* < 0.01 vs. NBP group: Mann–Whitney test between groups; # *p* < 0.05, ## *p* < 0.01, ### *p* < 0.001 vs. pre: Wilcoxon test between time.

**Table 4 ijerph-17-05910-t004:** Change of HDL function, antioxidant, cell repair and proinflammation markers after exercise.

Variable	NBP (*n* = 18)		HNBP (*n* = 12)		Source	*F*-Value	*p*-Value
	Pre	Post	Pre	Post			
***HDL function markers***							
SAA(ng/mL)	17.33	17.03	17.98	14.05 *^,##^	Time	3.909	0.058
	3.26	4.20	3.18	0.69	Groups	4.048	0.054
					T × G	2.872	0.101
SAA/HDL	0.32	0.32	0.37	0.25 ^*,###^	Time	11.459	0.002
	0.05	0.08	0.09	0.04	Groups	0.205	0.654
					T × G	9.881	0.004
APOA1 (ng/mL)	2182.20	3841.46 ^#^	2311.20	2769.78	Time	0.116	0.739
	1818.67	1971.41	1450.49	1297.58	Groups	0.141	0.713
					T × G	6.376	0.024
APOA1/HDL-C	27.33	51.67	84.24 **	55.19 ^#^	Time	0.023	0.881
	32.85	39.70	32.88	62.86	Groups	6.845	0.014
					T × G	2.921	0.098
***Antioxidant and cell repair markers***							
SOD2 (pg/mL)	50,126.30	56,463.16 ^##^	51,464.40	49,984.00	Time	2.863	0.102
	13,904.92	13,360.68	12,462.32	10,517.19	Groups	0.317	0.578
					T × G	7.418	0.011
HSP70 (ng/mL)	0.59	0.79 ^#^	0.49	0.73 ^###^	Time	12.721	0.001
	0.25	0.47	0.09	0.08	Groups	0.707	0.408
					T × G	0.129	0.722
***Pro-inflammation markers***							
TNF-α (pg/mL)	9.59	8.71	8.19	7.71 *	Time	2.334	0.138
	2.75	0.96	1.65	1.13	Groups	5.370	0.028
					T × G	0.204	0.655
CRP (mg/dL)	0.630	0.820	0.537	0.652	Time	1.754	0.196
	0.511	0.667	0.281	0.596	Groups	0.593	0.448
					T × G	0.106	0.748
IL-6 (pg/mL)	14.68	11.39	9.85	9.54 *	Time	2.427	0.130
	10.51	2.88	0.85	1.04	Groups	2.994	0.095
					T × G	1.659	0.208
***Anti-inflammation markers***							
IL-15 (pg/mL)	22.39	17.59	20.93	17.34	Time	4.110	0.052
	11.31	8.75	10.09	6.82	Groups	0.087	0.770
					T × G	0.086	0.772
IL-4 (pg/mL)	14.41	13.88	12.83 *	12.43 *^,#^	Time	1.535	0.226
	2.03	2.40	0.45	0.61	Groups	7.826	0.009
					T × G	0.032	0.860

Values are mean and standard deviation; NBP: normal blood pressure; HNBP: high-normal blood pressure; T: time; G: groups; SAA: serum amyloid A; APOA-1: apolipoprotein A-1; HDL-C: high density lipoprotein cholesterol; SOD2: superoxide dismutases 2; HSP70: hit shock protein 70; TNF-α: tumor necrosis factor α; CRP: C-reactive protein; IL-6: interleukin 6; IL-15: interleukin 15; IL-4: interleukin 4; T: time; G: groups; * *p* < 0.05, ** *p* < 0.01 vs. NBP group: Mann–Whitney test between groups; # *p* < 0.05, ## *p* < 0.01, ### *p* < 0.001, vs. pre: Wilcoxon test between time.

**Table 5 ijerph-17-05910-t005:** The 10-year cardiovascular disease risk according to blood pressure and lipids.

Variable	NBP (*n* = 18)		HNBP (*n* = 12)		Source	*F*-Value	*p*-Value
	Pre	Post	Pre	Post			
FRS Your Heart-10 Risk (%)	7.98	7.47	17.63 ***	16.17 ***^,#^	Time	4.361	0.046
	2.82	2.29	6.39	5.40	Groups	37.561	0.001
					T × G	1.018	0.322
Your Heart/Vascular Age (yr)	65.22	63.56	82.00 ***	81.67 ***	Time	0.479	0.494
	9.60	9.05	6.12	6.89	Groups	40.191	0.001
					T × G	0.213	0.648

Values are mean and standard deviation; NBP: normal blood pressure; HNBP: high-normal blood pressure; T: time; G: groups; FRS: Framingham risk score; T: time; G: groups; *** *p* < 0.001, vs. NBP group: Mann–Whitney test between groups, # *p* < 0.05, vs. pre: Wilcoxon test between time.

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
