# Peer review of "Can Active Aerobic Exercise Reduce the Risk of Cardiovascular Disease in Prehypertensive Elderly Women by Improving HDL Cholesterol and Inflammatory Markers?"

_ijerph, 2020, doi:10.3390/ijerph17165910_

Round 1
Reviewer 1 Report
It is a very interesting manuscript. It is well described, very well documented and has a good methodological design. Analyzes the functions measured both biochemical level, functional and body composition. However, the methodology should increase the information of the participating subjects in relation to whether or not they have other pathologies. The section referring to exercise intensity should also be clarified, as well as the statistical analysis (detailed below).
Methods
Page 3, line 111. Did the participating subjects also have other pathologies? or those with nothing but the alteration in blood pressure were selected. It would be interesting to further detail this information.
Page 4, line 139-142. In the manuscript it is described that the intensity was calculated from the Karvonen formula.
And how was the intensity of 65% FCmax controlled?
Did they use telemetry logging to monitor it?
If not, I don't see how you can control the intensity at just 65%.
However, if previously (in lines 125-127) it is described that the intensity of the exercise will be controlled from the RPE, then it is later said that the intensity of the aerobic exercise was 65% HRmax.
Or is the RPE just for resistance training work?
Was the intensity of exercise always 65% of HRmax during the 6 months of the exercise program?
If so, would it not have been good to increase the intensity of exercise, for example, 5% every 2 months until reaching 75% HRmax for example?
In other words, gradually increase the intensity of the exercise. In addition, it must be taken into account that the ventilatory threshold of trained older people is between 65-75% VO2max, which is approximately 75-85% FCmax.
All of the above should be clarified.
Page 5, line 188-195.
When the t-tests are performed, it is when the normality of the variables is met (not indicated in the statistical analysis).
Could you provide information on whether there is normal distribution?
In the event that any variable is not normal, the Wilcoxon and Mann-Whitney U tests should be performed, replacing the t-tests and leaving it indicated in the statistical analysis section.
Results
Page 5, line 199. The abbreviations LBM understand that it refers to Lean Body Mass? LBM does not appear earlier in the document. Not even when describing the variables to measure of body composition (lines 157-158).
Please review and add what the acronym means. Furthermore, it does not appear in Table 2.
Author Response
Response to Reviewer 1 Comments
It is a very interesting manuscript. It is well described, very well documented and has a good methodological design. Analyzes the functions measured both biochemical level, functional and body composition. However, the methodology should increase the information of the participating subjects in relation to whether or not they have other pathologies. The section referring to exercise intensity should also be clarified, as well as the statistical analysis (detailed below).
Methods
Point-1: Page 3, line 111. Did the participating subjects also have other pathologies? or those with nothing but the alteration in blood pressure were selected. It would be interesting to further detail this information.
Response-1 : We added the following sentence. ‘However, it was composed of subjects confirmed by the doctor's diagnosis as having no special diseases except blood pressure.’
Point-2 : Page 4, line 139-142. In the manuscript it is described that the intensity was calculated from the Karvonen formula.
And how was the intensity of 65% FCmax controlled?
Did they use telemetry logging to monitor it?
If not, I don't see how you can control the intensity at just 65%.
However, if previously (in lines 125-127) it is described that the intensity of the exercise will be controlled from the RPE, then it is later said that the intensity of the aerobic exercise was 65% HRmax.
Or is the RPE just for resistance training work?
Was the intensity of exercise always 65% of HRmax during the 6 months of the exercise program?
If so, would it not have been good to increase the intensity of exercise, for example, 5% every 2 months until reaching 75% HRmax for example?
In other words, gradually increase the intensity of the exercise. In addition, it must be taken into account that the ventilatory threshold of trained older people is between 65-75% VO2max, which is approximately 75-85% FCmax.
All of the above should be clarified.
Response-2 : We think it's the right intellectual. The following sentence has been added to complement it more specifically. ‘Adjustment of exercise intensity was applied in the range of 65-75% HRmax as a whole, and was maintained in the range of 65-70% HRmax in the first half and 70-75% HRmax in the second half. Control of this intensity was monitored using heart rate and RPE as described above.’
Point-3 : Page 5, line 188-195.
When the t-tests are performed, it is when the normality of the variables is met (not indicated in the statistical analysis).
Could you provide information on whether there is normal distribution?
In the event that any variable is not normal, the Wilcoxon and Mann-Whitney U tests should be performed, replacing the t-tests and leaving it indicated in the statistical analysis section.
Response-3 : It was modified as follows.
the independent t-test between groups by period and the paired t-test between periods by group.
- Mann-Whitney U test between groups by period and the Wilcoxon test between periods by group.
Results
Point-4 : Page 5, line 199. The abbreviations LBM understand that it refers to Lean Body Mass? LBM does not appear earlier in the document. Not even when describing the variables to measure of body composition (lines 157-158).
Please review and add what the acronym means. Furthermore, it does not appear in Table 2.
Response-4 : Next sentence was removed.
‘However, LBM was significantly lower in the HNBP group compared with the NBP group in both pre (p<0.05) and post (p<0.01) measurement in Table 2.’

Reviewer 2 Report
Dear authors,
the study is well-designed, interesting easy-to-apply intervention, and the manuscript is really well-written. However, I would suggest some clarifications before publication. Here are my concerns:
- First of all, I failed to find the reference for CONSORT reporting guidelines. It will help to improve readers' experience and report transparency.
- In the methods is clear that it is a RCT - should be stated in the title (see Consort guidelines);
- How and from where the subjects were recruited should be described;
- It feels like it is a small sample size for so many outcomes - I strongly recommend that authors describe the sample size calculations accordingly to primary outcomes;
- Still considering the sample size, is not clear if it is a pilot, a proof of concept trials or a full trial. It needs some clarification.
- The statistical analysis seems odd. Considering it is a RCT why the use of t-tests? Isn't better to keep the Anova 2-way for time, group and interaction alone? It adds confusion and do not add information to the study. I assume t-test was used to detect some differences that are underpowered for Anova 2-way analysis but it does not solve the problem of a small sample size. I suggest the authors to switch to effect sizes analysis, in addition to the Anova or search for some other alternative.
- There is a strong statement in the lines 203-204 that does not seem appropriate. I think the trial has issues with external validity at this point, and a small sample size for this kind of assumptions. I suggest to remove it from results and discuss limitations in the discussion section.
- My reporting and statistical suggestions might impact the presentation of the results and therefore the discussion, so at this point no comments about the discussion section.
- Consider adding a paragraph of "study limitations".
Author Response
Response-Reviewer-2 Comment
The study is well-designed, interesting easy-to-apply intervention, and the manuscript is really well-written. However, I would suggest some clarifications before publication. Here are my concerns:
Point 1 : First of all, I failed to find the reference for CONSORT reporting guidelines. It will help to improve readers' experience and report transparency. In the methods is clear that it is a RCT - should be stated in the title (see Consort guidelines);
Response 1 : The following sentence was added to supplement the CONSORT reporting guidelines in methods. ‘The study subjects were selected from among the members of the Elderly Health Promotion Center (Out of 1,000 members) in Daegu, South Korea, who viewed the notice and voluntarily agreed to participate.’
Point 2 : How and from where the subjects were recruited should be described; It feels like it is a small sample size for so many outcomes - I strongly recommend that authors describe the sample size calculations accordingly to primary outcomes; Still considering the sample size, is not clear if it is a pilot, a proof of concept trials or a full trial. It needs some clarification.
Responses 2 : It is considered to be an appropriate intellectual, and considering this, the following sentence is added. ‘In the process of selecting candidates among subjects who were allowed to participate through doctor's diagnosis and randomly classifying them into two groups, the limitation of the sample size was considered as the limitation of this study.’
Point 3 : The statistical analysis seems odd. Considering it is a RCT why the use of t-tests? Isn't better to keep the Anova 2-way for time, group and interaction alone? It adds confusion and do not add information to the study. I assume t-test was used to detect some differences that are underpowered for Anova 2-way analysis but it does not solve the problem of a small sample size. I suggest the authors to switch to effect sizes analysis, in addition to the Anova or search for some other alternative.
Response 3: : It was modified as follows.
the independent t-test between groups by period and the paired t-test between periods by group.
- Mann-Whitney U test between groups by period and the Wilcoxon test between periods by group.
Point 4 : There is a strong statement in the lines 203-204 that does not seem appropriate. I think the trial has issues with external validity at this point, and a small sample size for this kind of assumptions. I suggest to remove it from results and discuss limitations in the discussion section. My reporting and statistical suggestions might impact the presentation of the results and therefore the discussion, so at this point no comments about the discussion section.
Response 4 : We deleted the following sentence in in the lines 203-204
‘This allows the assumption that exercise has no effect on obesity reduction in HNBP patients. In both groups, the mean WHR decreased significantly (p<0.05) after the exercise program.’
Point 5 : Consider adding a paragraph of "study limitations".
Response 5 : I added the following sentence in the discussion
‘However, it is considered that this study is limited to the elderly in a limited area and the application of the results is very limited considering the very small sample size, and further studies including more subjects are required.’

Reviewer 3 Report
The work presented is in an area of ​​great interest since heartdisease represents a major expense for global health The care of the third age related to inflammatory parameters of arteries and overweight are of great interest in developed countries The results are interesting, well used in the discussion and reinforce the authors' conclusions
Author Response
Response-Review-3-Comments
The work presented is in an area of ​​great interest since heart disease represents a major expense for global health. The care of the third age related to inflammatory parameters of arteries and overweight are of great interest in developed countries. The results are interesting, well used in the discussion and reinforce the authors' conclusions.
Response : Thank you very much.

Round 2
Reviewer 2 Report
Dear authors,
although some improvement can be seeing in your manuscript, still does not make improvements in the reader experience. Some highlights:
- Results discussion are not clear to which statistical analysis is referring to. I will recommend again some different approach, as described in the previous review;
- As an example: authors suggest in lines 310-312 some benefits, that are not in agreement to statistical results.
- Also, authors refer to physical fitness benefits (line 274) that do not agree with statistical significance presented for ANOVA.
- Effect sizes analysis would help solving these problems.
- T-test analysis for groups is unclear: is it considering both time-points together or separately?
- The incorporation of a non-parametric test in the methods does not reflect the results - I was not able to localize these analysis.
- Usually Mann–Whitney–Wilcoxon is one test - not 2 as described.
- Adding a new analysis does not improve the fact that the statistics is already unclear in the manuscript.
- The study still do not incorporate CONSORT guidelines;
- The manuscript would also benefit from English and plain language review: some sentences are too long, and minor misuse of prepositions would need some attention.
Author Response
Response to Reviewer 2 Comments
Point 1 : Results discussion are not clear to which statistical analysis is referring to. I will recommend again some different approach, as described in the previous review; As an example: authors suggest in lines 310-312 some benefits, that are not in agreement to statistical results. Also, authors refer to physical fitness benefits (line 274) that do not agree with statistical significance presented for ANOVA. Effect sizes analysis would help solving these problems.
Response 1 : The following sentence was added.
‘It can be said that these differences between groups are difficult to accept as clear results. Therefore, it is thought that continuous research is required to further supplement this part.’
Point 2 : T-test analysis for groups is unclear: is it considering both time-points together or separately? The incorporation of a non-parametric test in the methods does not reflect the results - I was not able to localize these analysis. Usually Mann–Whitney–Wilcoxon is one test - not 2 as described. Adding a new analysis does not improve the fact that the statistics is already unclear in the manuscript.
Response 2 : The following partial correction was performed in table.
* p<0.05, ** p<0.01, *** p<0.001 vs NBP group: Mann-Whitney test between groups between groups; # p<0.05, ## p<0.01, ### p<0.001 vs pre: Wilcoxon test between time.
Point 3 : The study still do not incorporate CONSORT guidelines;
Response 3 : The following sentence was added.
‘Therefore, the limitation of this study is that the random selection method cannot be considered to be applied in the process of selecting a perfect subject.’
Point 4 : The manuscript would also benefit from English and plain language review: some sentences are too long, and minor misuse of prepositions would need some attention.
Response 4 : Some contents have been modified and supplemented.